# Development and Characterization of Unmodified and Modified Natural Rubber Composites Filled with Modified Clay

**DOI:** 10.3390/polym14173515

**Published:** 2022-08-27

**Authors:** Adisak Keereerak, Nusara Sukkhata, Nussana Lehman, Yeampon Nakaramontri, Karnda Sengloyluan, Jobish Johns, Ekwipoo Kalkornsurapranee

**Affiliations:** 1Division of Physical Science, Faculty of Science, Prince of Songkla University, Hat-Yai, Songkhla 90110, Thailand; 2Rubber Technology and Engineering Program, International College, Prince of Songkla University, Songkhla 90110, Thailand; 3Sustainable Polymer & Innovative Composite Materials Research Group, Department of Chemistry, Faculty of Science, King Mongkut’s University of Technology Thonburi, Bangkok 10140, Thailand; 4Department of Physics, Rajarajeswari College of Engineering, Bangalore 560074, India

**Keywords:** bentonite clay, intercalation, ultrasonic method, natural rubber, epoxidized natural rubber

## Abstract

Novel composite based on rubber and modified bentonite clay (Clay) was investigated. The modified bentonite clay was developed by dispersing in ethanol solutions (Et-OH) using ultrasonic method. The effect of Et-OH on the dispersion of bentonite clay at various mixing temperatures in case of different type of rubber matrix, i.e., natural rubber (NR), epoxidized natural rubber (ENR25, ENR50) on dynamic mechanical rheology, Payne effect, XRD and mechanical properties of rubber composites were studied in detail. The bentonite clay dispersion in Et-OH at a mixing temperature of 80 °C improves the intercalation and exfoliation in rubber chains. Bentonite clay is highly intercalated in ENR 50-Clay composite, which can be confirmed from its superior mechanical properties. The results indicated that sonication of bentonite clay in Et-OH improves the interlayer spacing of bentonite clay by partial intercalation of rubber matrix.

## 1. Introduction

Composite materials are the combination of two or more phases of materials with different properties such that the combinations provide a product with the intermediate characteristics of the components [1]. Generally, the properties of composite materials are influenced by the two materials involved as well as the method of processing [2]. A composite consists of three components: (i) the matrix as the continuous phase; (ii) the reinforcements as the discontinuous or dispersed phase, including fiber and particles; and (iii) the fine interphase region, also known as the interface [3]. Different phases in the composite are combined together to obtain a composite with unique properties. Rubber composites are the typical examples, since curing is a recipe of chemicals for the modification of its elasticity and strength [1]. Natural rubber (NR) is a biopolymer based on *cis*-1,4-polyisoprene with excellent elastic properties, resilience and damping behavior. However, the limitations of NR are poor chemical resistance and processing ability. The applications of NR including tires, machinery components, gloves, toys, shoe soles, elastic bands, flippers, and sport equipment [1,4,5,6].

Reinforcement of filler particles into NR is a promising technique that affords to enhance the performance of NR. For instance, clay silicates in the form of montmorillonite, hectorite, bentonite, etc. are used directly or modified before adding into the rubber matrix. This filler has drawn a great attention when layered silicates are dispersed in the polymer matrix at the nanoscale level and the reinforcement ability can be improved by replacing the inorganic exchange cations in the surface of the clay by alkyl ammonium surfactants [7]. Bentonites essentially consist of clay minerals of smectite groups. It has been widely used as industrial raw materials for many industries. The properties of natural bentonites consist of 2:1 layered aluminosilicate and negative layers in bentonite balance electrically by equally charged cations. The clay layers are negatively charged as a consequence of an isomorphic distribution of exchangeable counterbalanced cations such as Na^+^ and Ca^2+^ in the interlayer. As a result of hydration of inorganic cations on the exchange site, the clay mineral surface is hydrophilic [8]. It leads them not so suitable to mix and interact with most of the polymer matrices with higher hydrophobicity. Therefore, organically modified layered silicates can be generated by modifying clays with various organic surfactants through intercalation process. Ion exchange is an easy and traditional method to modify the surface of clay. However, the disadvantages of this method including the cations are not strongly bound to the clay surface and also small cation molecules can substitute the cations present in the clay surface [9,10]. The interlayer spacing contains exchangeable cations that can be replaced by organic cationic surfactants (e.g., alkylammonium or alkylphosphonium cations) to improve the compatibility of clay with rubber. This method improves the interlayer distances of montmorillonite and exhibits organophilic and hydrophobic nature on its surface, and the treatment is usually carried out in aqueous or aqueous-ethanol media [11,12].

The clay layer can be steadily dispersed in water for the hydration of ion among the layers; the layers are separated from each other. Some polar compounds can intercalate to the clay layer galleries. These characteristics provide the way to prepare the rubber-clay composites [13]. Intercalation techniques are helpful to insert organic molecule into clay gallery, effectively. Increase of interlayer space in clay particles is the most important factor to disperse clay in rubber matrix [14,15].

The main objective of this work is to prepare the nanocomposites of NR and bentonites clay by using ethanol as a dispersing agent with the help of ultrasonicator. This study is also aimed to separate clay layers by intercalation of bentonite clay in ethanol (Et-OH) solution, which enhances the dispersion ability of bentonite clay (dispersed phase) in rubber network microstructure (rubber matrix phase). Initially, the effect of mixing temperature of NR compounding (60 °C, 80 °C) without and with Et-OH dispersed bentonite clay was studied. Secondly, the effect of rubber types including natural rubber or epoxidized natural rubber (ENR) with 25% epoxy group (ENR-25) and 50% epoxy group (ENR-50) was also studied. The characteristics of dynamic properties, the Payne effect, XRD and mechanical properties have been analyzed.

## 2. Materials and Methods

### 2.1. Material

Natural rubber (Standard Thai Rubber 5L grade) with density of 0.92 g/cm^3^ was manufactured by Chalong Latex Industry Co., Ltd., Songkhla, Thailand. Epoxidized natural rubber containing 25 mole% and 50 mole% epoxide units was obtained from Muang Mai Guthrie PCL, Phuket, Thailand. The raw bentonite clay was collected from Siam Chemicals Co., Ltd., Samudprakarn, Thailand. Ethanol solution was used as a dispersing agent for the preparation of organically modified layered silicates. Compounding ingredients including zinc oxide (ZnO) and stearic acid were purchased from Boss Optical Limited Partnership, Songkhla, Thailand. 2-2′-Dithiobis (benzothiazole) or MBTS and sulfur were supplied by Siam Chemicals Co., Ltd., Samudprakarn, Thailand.

### 2.2. Preparation of Rubber–Clay Composites

The bentonite clay was dispersed in 97 wt% of ethanol solution. The ratio of bentonite clay to ethanol was 1:2 (weight/weight). The dispersions were sonicated for 2 min with a QSONIC Q700 sonicator processor of 700 W output at 80 Hz.

Rubber-clay composites were prepared through melt blending method in a laboratory mixer of brabender type with chamber volume of 430 cm^3^ at 60 rpm. Temperature of the mixing chamber was fixed at 60 °C and 80 °C for 8 min. The ingredients and their amounts of compounds are as shown in Table 1. The rubber compounds were vulcanized through a compression molding machine at 150 °C to produce NR sheets with dimensions of 125 × 125 mm^2^. The sample prepared without bentonite clay has been named as NR.

### 2.3. Characterization of Rubber-Clay Composites

#### 2.3.1. Dynamic Mechanical Rheology and Payne Effect

Dynamic mechanical behavior and Payne effect were measured using a rubber process analyzer (RPA, Alpha Technologies, Hudson, OH, USA). An uncured sample was first vulcanized directly in the RPA at 150 °C for its respective cure time or vulcanized time (Tc_95_) and subsequently cooled down to 100 °C prior to Payne effect test. The test was set with a strain sweep in the range of 0.56–100% at a frequency of 1.00 Hz. The difference of storage modulus at 0.56% and 100% strain has been used to calculate Payne effect.

#### 2.3.2. X-ray Diffraction

The interplanar spacing of clay was monitored using an X-ray diffractometer, Empyrean, PANalytical, Netherlands with Cu-Kα radiation at 40 kV and 30 mA and wavelength of 0.154 nm at room temperature. As an empirical equation, Bragg’s law was used to compute the crystallographic spacing (d-spacing) or the interplanar spacing in composites. The angle 2θ was varied in the range of 3–90° with a scanning rate of 2° min^−1^.

#### 2.3.3. Mechanical Properties

Mechanical properties of the resulting composites were investigated using an Instron machine (Model 3365) according to ASTM D412. The specimens were cut into dumbbell shape and the test was conducted with a load cell of 1000 N at a cross head speed of 500 mm/min.

## 3. Results and Discussion

### 3.1. Effect of Et-OH on Bentonite Clay Dispersion in NR and NR-Clay Composites

For a better understanding of dispersion of bentonite clay in NR and NR-Clay composites, rheological parameters of the prepared composites are determined from shear thinning and elastic behavior [16]. NR without bentonite clay at the mixing temperature of 60 °C, NR-Clay composite with bentonite clay without dispersed in Et-OH at various mixing temperatures (60 °C, 80 °C) and NR-Clay composite with bentonite clay dispersed in Et-OH at the mixing temperature of 80 °C are studied.

Basically, elastic modulus (G′) is the measure of elastic behavior (stored and recovered energy) in a cyclic deformation [16]. Figure 1 shows the storage modulus against the percentage of strain in the range of 0.56–100%. On increasing the %strain, a gradual reduction in storage modulus is noticed due to the breakdown of filler-filler networks. The final storage modulus is still high at the maximum %strain represents the combination of filler structure in NR molecule and the filler polymer interaction [17]. Moreover, NR-Clay composite prepared by dispersing bentonite clay in Et-OH at the mixing temperature of 80 °C exhibited the lower value of storage modulus than the other NR-Clay composites. Therefore, it makes favorable to the rubber chain movement by increasing the viscous behavior that can be represented as “sliding effect” rather than a “reinforcing effect” [16].

Figure 2 displays the Payne effect of various NR-Clay composites. It can be seen that the Payne effect of NR-Clay composite prepared by reinforcing bentonite clay through the dispersion in Et-OH at the mixing temperature of 80 °C is 26.05. This value is lower than the NR-Clay composites without dispersing bentonite clay in Et-OH (at both the mixing temperatures of 60 °C and 80 °C). Payne effect is related to the breakdown of filler network or filler agglomerated structures from filler-filler interactions [18]. In case of lower value, the Payne effect can be explained due to lower agglomeration of fillers in the composite. It clearly indicates that, the NR-Clay composite reinforced with Et-OH dispersed bentonite clay at a mixing temperature of 80 °C exhibited a lower agglomeration with better particle distribution in Et-OH. Due to the higher mixing temperature and dispersion in Et-OH produces better shear force of mixing between NR and filler by decreasing the interaction between positive ion and silicon structure. This behavior might exhibit breakdown and more slip of bentonite clay planes which results in the increased dispersion in rubber matrix.

Figure 3 illustrates the XRD patterns of NR and NR-Clay composites with and without dispersing bentonite clay in Et-OH solution at different mixing temperatures. Pure NR shows clearly the amorphous characteristic as it does not have any crystalline peaks. Normally, the diffraction peaks can be seen at the angles 5.71°, 7.61°, 12.55°, 19.89°, 20.53°, 21.03°, 21.42°, 25.09°, 35.05°, 35.92° and 61.69° [19]. The NR-Clay composites prepared at the mixing temperature of 60 °C exhibits the characteristic peaks at the angle of diffraction 2θ at 11.28°, 22.86°, 27.86°and 28.81°. The NR-Clay composites developed at the mixing temperature of 80 °C show some sharp peaks at 10.12°, 22.60°, 26.88° and 28.33°. The diffraction angle is related to the characteristic peaks of bentonite clay at 21.42° (110), 27.20° (210) and 30.01° (124) [19,20,21]. Moreover, the XRD patterns indicated that the NR-Clay composites consist of Heulandite-Ca and Montmorillonite-15A as the major and minor constituents.

In addition, the angle of diffraction displaced towards lower value upon adding modified bentonite clay in the composites. Especially, the angle reduces from 19.94° to 18.15° that shows the increment of inter-planar spacing of 3.85 to 4.99 nm. From these results, the d-spacing of NR-Clay composites with bentonite clay in Et-OH and temperature of 80 °C provides the highest value among the other NR-Clay composites as shown in Table 2. This is the reason to exhibit better homogeneous dispersion of bentonite clay in NR-Clay composite using bentonite clay dispersed in Et-OH before mixing in internal mixer. Moreover, the increasing of inter-planar spacing reveals that the intercalation of NR molecular chains in between the layers of bentonite clay. It is an indication of enhanced mechanical properties compared to the NR-Clay composites prepared without dispersing in Et-OH. Moreover, these results are related to the values of Payne effect, and it can be explained in terms of lower filler-filler interaction.

As discussed in the previous results of Payne effect and the XRD analysis of NR-Clay composite asserted well dispersion of bentonite clay in Et-OH and mixed in internal mixer at temperature of 80 °C. These properties are better than the other NR and NR-Clay composites. Here, the variation in mechanical properties of these composites is also confirmed from this phenomenon as investigated.

Figure 4 shows the stress-strain curves of NR and NR-Clay composites prepared at various conditions. An increment of stress at lower %strain is due to the entanglement of molecular chains. Furthermore, on increasing the %strain, stress is found to be decreased as a result of its easily movement, breakdown and scission in their molecular chains. Figure 5 illustrates the tensile strength of NR and NR-Clay composites at various conditions. There is no significant difference in the tensile strength, 100% modulus and 300% modulus in case of NR-Clay prepared by dispersing bentonite clay in Et-OH but shows a slight increase when compared to the other NR-Clay composites. This might be due to the result of rubber without adding any curing agents during compounding.

Figure 6 and Table 3 displays the tensile properties of vulcanized NR and NR-Clay composites. The tensile strength values of vulcanized NR, vulcanized NR-Clay composite, and vulcanized NR-Clay composite using Et-OH dispersed bentonite clay are 2.92, 3.87, and 4.92 MPa, respectively. Moreover, 100% and 300% moduli are increased for the composites prepared at higher temperature and dispersed in Et-OH conditions. This is attributed to the better dispersion of filler in rubber matrix resulted in the highest value in case of clay dispersed in Et-OH. This behavior can be related to the above results in terms of proper dispersion of bentonite clay in the rubber matrix. In addition, the elongation at break is increased from 578% to 656% as changing the conditions with Et-OH dispersed bentonite clay. Due to the occurrence of easy slip of NR molecular chains from the bentonite clay planes, which increases the surface area of bentonite clay.

The possible dispersion model of bentonite clay dispersed and undispersed in Et-OH is shown in Figure 7. The process with dispersed bentonite clay in Et-OH exhibits better dispersion in NR matrix. This might be due to the interaction (hydrogen intramolecular force) between hydroxyl group (-OH) on the surface of bentonite clay and hydroxyl group of Et-OH. It increases the sliding of bentonite clay plane tends to lower agglomeration among the fillers and properly dispersed in rubber matrix. The NR matrix is more intercalated and exfoliated in bentonite clay as confirmed from the XRD analysis (increasing of d-spacing).

### 3.2. Effect of Rubber Types in NR-Clay Composites on the Properties of NR/Clay Composites

From the above discussion, the optimum condition for the preparation of NR-Clay composite is the one with bentonite clay of 20 phr dispersed in Et-OH at a mixing temperature of 80 °C. In this section, the types of rubber (NR and modified NR) including NR, ENR-25 and ENR-50 are optimized.

Figure 8 shows the variation of storage modulus against the percentage of strain in the range of 3–40% of NR-Clay, ENR25-Clay and ENR50-Clay composites. The storage modulus is slightly decreased upon increasing the %strain up to 10% and further a sharp reduction can be seen until 100%. This behavior is mainly due to the existence of filler network at lower strain and breakdown occurs at higher %strain [21]. The storage modulus of ENR50-Clay composite is higher than NR-Clay and ENR25-Clay. It indicates the stronger structure of filler network and stronger filler-rubber interaction [5,22].

Figure 9 shows the XRD patterns of NR-Clay, ENR25-Clay and ENR50-Clay composites. The ENR25-Clayet composites exhibited peaks at the diffraction angles 22.94°, 26.8° and 28.13°. Pointed peaks are observed at 22.71°, 26.42° and 28.15° in case of ENR50-Clayet composites. In cases of NR-Clay, ENR25-Clay and ENR50-Clay composites, the broad peak in the range of 12–24 degree shifts to lower side. Furthermore, the angles slide to lower the angle from 19.94, 19.91 and 19.42, respectively. Additionally, the ENR50-Clay composite shows the displacement of peaks at 10.03° and 29°. This is attributed to the impregnation of elastomeric phase into the silicate galleries, hiding the regular stacked-layer structure of the modified bentonite clays and extends to an intercalated structure. Moreover, intensity of the sharp peak (characteristic peak) is also higher on increasing the epoxy degree in ENR composites. Dominantly, the intensity of diffraction peak at 10.03° is slightly higher than 9.99 and 9.9°. The d-spacing is found to be increased from 3.87 to 4.92 nm at 19.98°, 19.91° and 19.20° as shown in Table 4. It exhibits the intercalation of rubber molecules in between the layers of bentonite clay. Furthermore, the polarity of the modified NR (ENR) supports intercalation of elastomer chain into the modified bentonite clay leads to better intercalation and dispersion in the matrix [23,24].

Figure 10 and Table 5 show the tensile properties of NR-Clay, ENR25-Clay and ENR50-Clay composites. Modified NR-Clay composites exhibited higher tensile strength than unmodified NR-Clay composite. Among the modified NR-Clay composites, ENR50-Clay composite becomes mechanically harder when compared to the composite of ENR25-Clay about 2 times. This phenomenon is due to the increased epoxy content that leads to enhance the property (an increase of T_g_ at higher epoxy content) in ENR matrix [25]. The Tg of ENR25 and ENR50 are −47.5 °C and −30 °C, respectively. Moreover, the Tg of ENR increases linearly with the degree of epoxidation or epoxy content [24]. From previous research, Yokkhun et al. (2014) studied the preparation and characterization of ENR/modified montmorillonite clay (OC-MMT) composite. The Tg was found to be increased from −28 °C up to 18 °C along with epoxy content (10–50%). Increment of tensile properties was observed as a result of strong interaction between ENR and OC-MMT and also the degree of strain-induced crystallinity in ENR matrix [25]. Additionally, the presence of higher epoxy content will have a better interaction with the positive ion in bentonite clay. The greater number of Na^+^ coordination develops a stronger ion dipole interaction. Typically, addition of ENR with higher epoxy level in ENR-Clay composite increases the polarity or reactive functional groups. Bentonite clay also has the polarity in its structure and the compatibility between NR and clay is increased. It reflects in the higher tensile strength. Moreover, reinforcing index is the reason for the increasing trend of storage modulus curve in dynamic mechanical property.

As discussed in the previous sections, ENR50/Clay composite exhibits better properties in terms of Payne effect, XRD analysis and tensile properties when compared to ENR25/Clay and NR/Clay composites. This is attributed to the better dispersion of bentonite clay in the rubber matrix obtained by dispersing bentonite clay in Et-OH solution at a mixing temperature of 80 °C.

The possible interaction mechanisms between ENR (ENR-25, ENR-50) and bentonite clay are presented in Figure 11. Van der Waals intermolecular force between the H atom (positive cloud) of bentonite clay surface and O atom (negative cloud) on the epoxy ring group is the one type of interaction. The second type of interaction is ion dipole intermolecular force between Na^+^ ions (positive ion) of bentonite clay structure and O atom (negative cloud) of epoxy ring group. Finally, Covalent bonding between hydroxyl group (-OH) of bentonite clay surface and C atom at the backbone (neighbored O atom) on the epoxy ring functional group by deprotonating an atom and ring opening of these functional groups take place. Covalent bonding has been adapted from Tan et al. (2016) [26], as the occurrence of reaction between epoxy ring of ENR and Si–OH of Na-MMT surface. The increased interplanar spacing is also resulted in intercalation as seen in the XRD analysis.

## 4. Conclusions

Rubber composites based on modified bentonite clay dispersed in Et-OH solution by ultrasonic method were successfully prepared. The NR-Clay composite developed by using bentonite clay of 20 phr dispersed in Et-OH at a mixing temperature of 80 °C exhibited better properties in terms of Payne effect, XRD analysis and mechanical properties when compared to the composites of unmodified bentonite clay. Based on the effect of rubber types, XRD analysis showed better dispersion in ENR50-Clay composites rather than the other rubber composites. An increase in d-spacing (3.87 to 4.92 nm) and the shift of angle of diffraction to lower angle (19.98 to 19.20 degree) can also be seen from XRD analysis. Moreover, ENR50-Clay composite exhibited better tensile properties than the composites of NR-Clay and also exhibited two times higher than that of ENR25-Clay composite. As a result of increased epoxy content in ENR, the property of ENR matrix was also simultaneously enhanced. On evaluating the performance of composites, ENR50-Clay composite with modified bentonite clay dispersed in Et-OH solution improved the interlayer spacing of bentonite clay (well dispersion) by partial intercalation of rubber matrix.

## Figures and Tables

**Figure 1 polymers-14-03515-f001:**
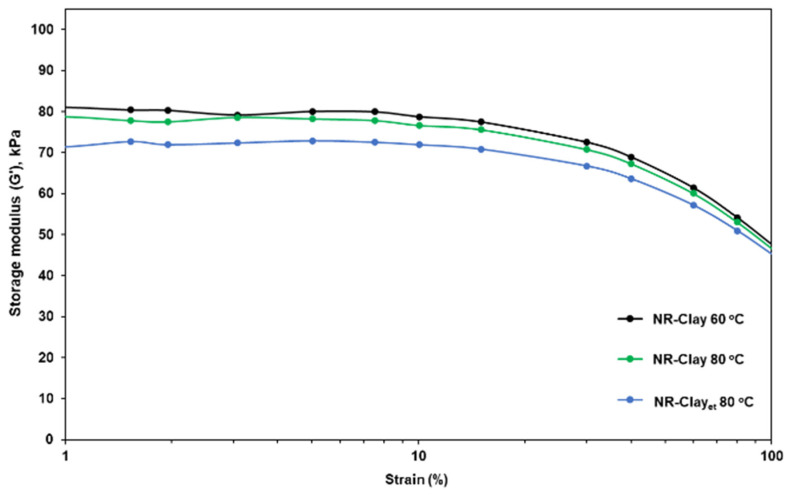
Storage modulus (G′) against %strain (1–100%) of NR, NR-Clay composites at the various mixing temperature (60 °C, 80 °C) and NR-Clay composites using Et-OH dispersed bentonite clay.

**Figure 2 polymers-14-03515-f002:**
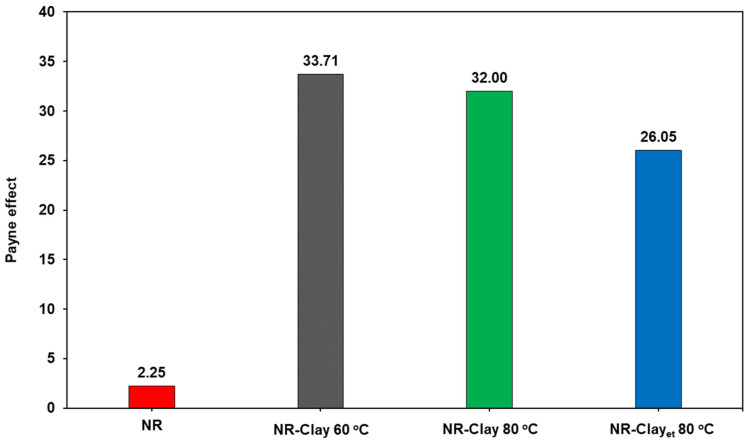
The Payne effect of NR, NR-Clay composites at various mixing temperatures (60 °C, 80 °C) and NR-Clay composites using Et-OH dispersed bentonite clay.

**Figure 3 polymers-14-03515-f003:**
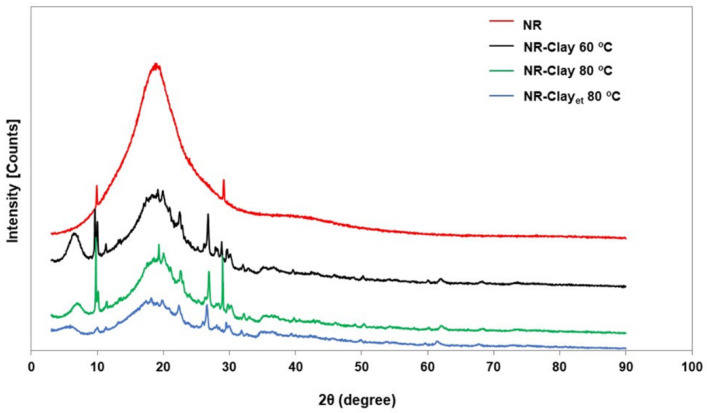
XRD pattern of NR, NR-Clay composites at various mixing temperatures (60 °C, 80 °C) and NR-Clay composites using Et-OH dispersed bentonite clay.

**Figure 4 polymers-14-03515-f004:**
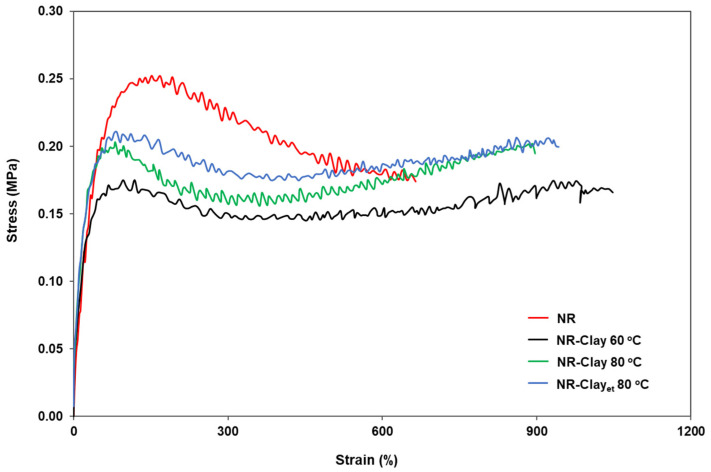
Stress-strain curve of NR, NR-Clay composites at various mixing temperature (60 °C, 80 °C) and NR-Clay composites using Et-OH dispersed bentonite clay.

**Figure 5 polymers-14-03515-f005:**
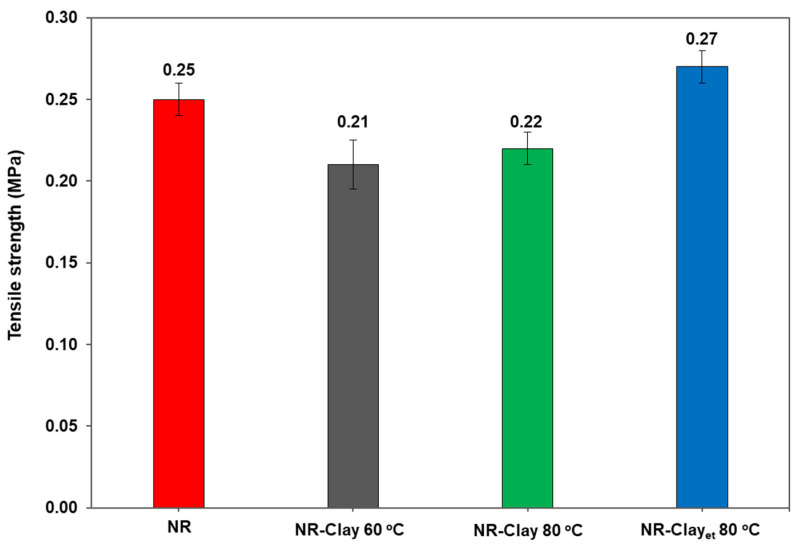
Tensile strength of NR, NR-Clay composites at various mixing temperatures (60 °C, 80 °C) and NR-Clay composites using Et-OH dispersed bentonite clay.

**Figure 6 polymers-14-03515-f006:**
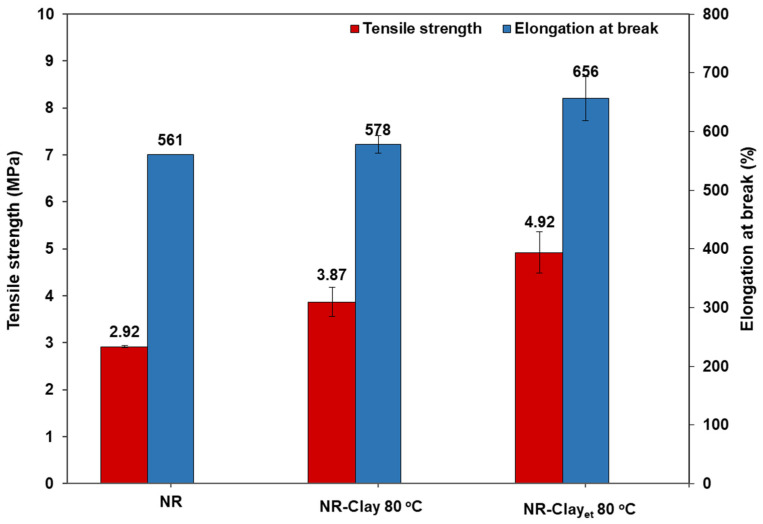
Tensile strength and elongation at break of vulcanized NR, NR-Clay composites at various mixing temperatures (60 °C, 80 °C) and NR-Clay composites of bentonite clay dispersed in Et-OH.

**Figure 7 polymers-14-03515-f007:**
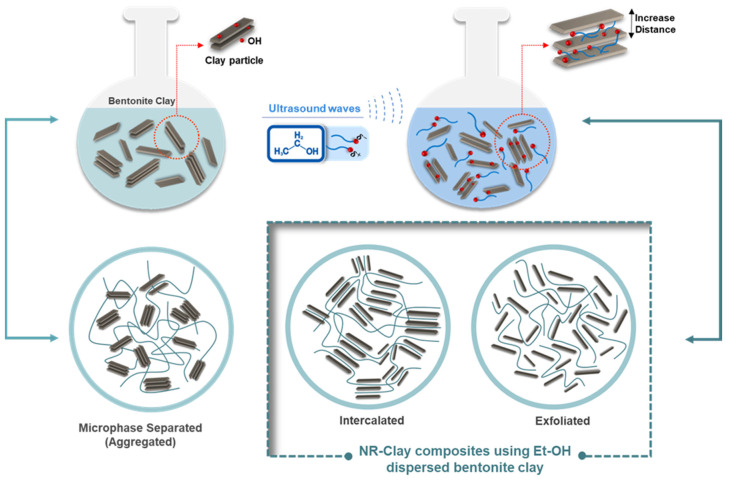
The proposed model and interaction between NR and bentonite clay in NR/Clay composites.

**Figure 8 polymers-14-03515-f008:**
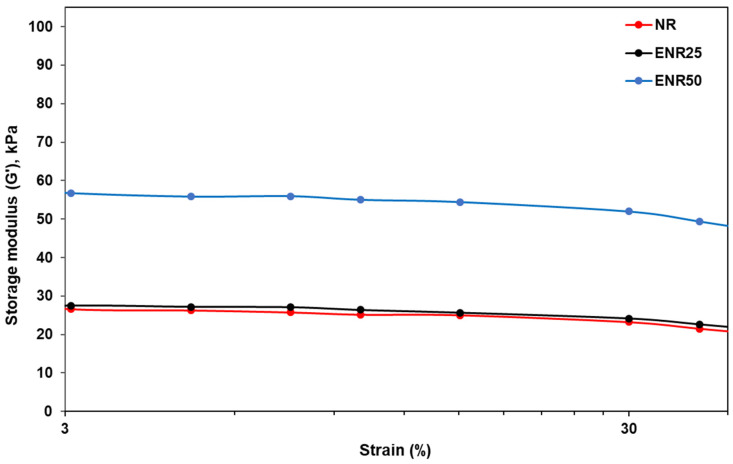
Storage modulus (G’) against %strain (3–40%) of NR-Clay, ENR25-Clay and ENR50-Clay composites at the mixing temperature of 80 °C prepared by dispersing bentonite in Et-OH solution.

**Figure 9 polymers-14-03515-f009:**
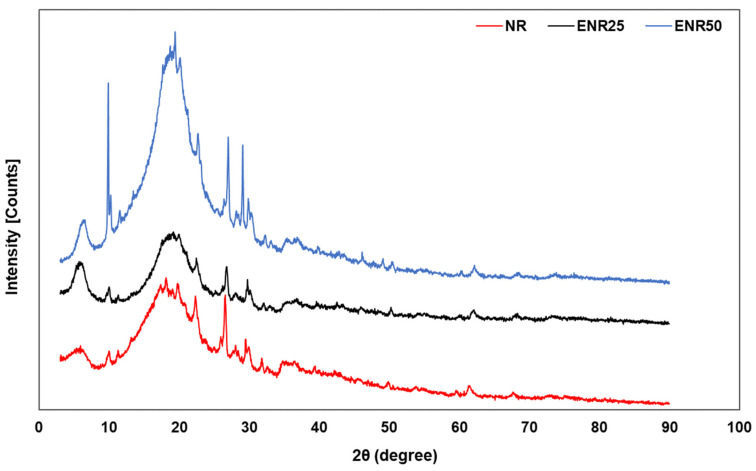
XRD patterns of NR-Clay, ENR25-Clay and ENR50-Clay composites at the mixing temperature of 80 °C by using dispersed bentonite clay in Et-OH.

**Figure 10 polymers-14-03515-f010:**
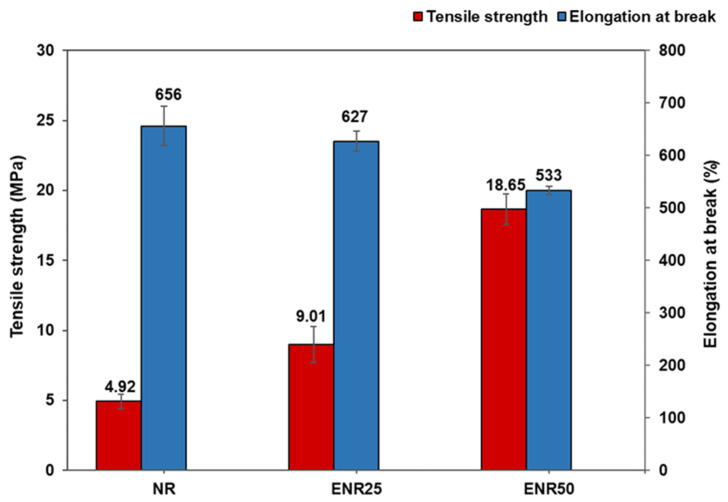
Tensile strength and elongation at break of vulcanized NR-Clay, ENR25-Clay and ENR50-Clay composites prepared at the mixing temperature of 80 °C and Et-OH used to disperse bentonite clay.

**Figure 11 polymers-14-03515-f011:**
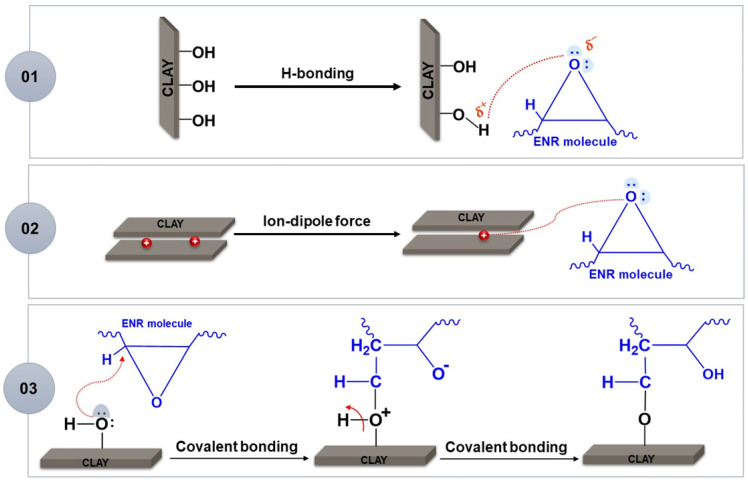
The proposed model and interaction of ENR and bentonite clay in ENR/Clay composites.

**Table 1 polymers-14-03515-t001:** Formulation of the rubber compounds.

Ingredients	Function	Designation			
		NR	NR-Clay	ENR25-Clay	ENR50-Clay
		Content (phr ^1^)			
^2^ NR	Matrix	100	100	-	-
^3^ ENR25	Matrix	-	-	100	-
^4^ ENR50	Matrix	-	-	-	100
Bentonite clay (dispersion in Et-OH)	Dispersion agent	0	20	20	20
Zinc oxide	Activator	5	5	5	5
Stearic acid	Activator	1	1	1	1
^5^ MBTS	Accelerator	1	1	1	1
Sulfur	Crosslinking agent	2.5	2.5	2.5	2.5

^1^ phr: parts per hundred parts of rubber. ^2^ NR: natural rubber. ^3^ ENR25: epoxidized natural rubber (25 mole% epoxide units). ^4^ ENR50: epoxidized natural rubber (50 mole% epoxide units). ^5^ MBTS: 2-2′-Dithiobis (benzothiazole).

**Table 2 polymers-14-03515-t002:** Two theta and d-spacing of NR/Clay composite is obtained from XRD analysis.

NR-Clay Composite	Two Theta (Degree)	d-Spacing (nm)
NR	-	-
NR-Clay 60 °C	11.28	7.84
	19.94	3.85
	22.86	3.89
	27.86	3.20
	28.81	3.10
NR-Clay 80 °C	10.12	8.74
	19.29	4.20
	22.60	3.93
	26.88	3.31
	28.33	3.15
NR-Clay_et_ 80 °C	10.03	8.82
	18.15	4.99
	22.67	3.92
	26.52	3.36
	28.06	3.18

**Table 3 polymers-14-03515-t003:** Mechanical properties of NR/Clay composites without curing agent.

Properties	NR	NR-Clay 80 °C	NR-Clay_et_ 80 °C
100% modulus (MPa)	0.46 ± 0.01	0.47 ± 0.02	0.45 ± 0.08
300% modulus (MPa)	0.97 ± 0.03	1.00 ± 0.02	1.12 ± 0.03
Tensile strength (MPa)	2.92 ± 0.02	3.87 ± 0.31	4.92 ± 0.44

**Table 4 polymers-14-03515-t004:** Two theta and d-spacing of Rubber-Clay composite obtained from XRD analysis.

Rubber-Clay Composite	Two Theta (Degree)	d-Spacing (nm)
NR	19.98	3.87
ENR25	19.91	4.46
ENR50	19.20	4.92

**Table 5 polymers-14-03515-t005:** Mechanical properties of NR-Clay, ENR25-Clay and ENR50-Clay composites.

Properties	NR	ENR25	ENR50
100% modulus (MPa)	0.45 ± 0.08	0.90 ± 0.34	1.41 ± 0.03
300% modulus (MPa)	1.12 ± 0.03	1.56 ± 0.02	4.05 ± 0.01
Reinforcing index (MPa)	2.49 ± 0.37	1.73 ± 0.06	4.05 ± 0.33
Tensile strength (MPa)	4.92 ± 0.54	9.01 ± 1.28	18.65 ± 1.08

## Data Availability

Not applicable.

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
