# Peer review of "Development and Characterization of Unmodified and Modified Natural Rubber Composites Filled with Modified Clay"

_polymers, 2022, doi:10.3390/polym14173515_

Round 1
Reviewer 1 Report
In this work, the effect of Et-OH on the dispersion of bentonite clay at various mixing temperatures in case of different type of rubber matrix on dynamic mechanical rheology, Payne effect, XRD and mechanical properties of rubber composites were studied in detail. I think it can be accept after minor revise.
(1) Line 13. “Novel composites based on rubber and modified bentonite clay (Clay) with enhanced dispersion in ethanol solutions (Et-OH) using ultrasonic method was investigated.” This statement is ambiguous, please revise it.
(2) I think the author's introduction is not comprehensive in explaining why the intercalation method is used, the role of ethanol in it, and the previous research progress.
(3) I am very confused about the design of Table 1. I think the content of each component should be stated more clearly. Moreover, I suggest the author define the name of the sample clearly.
(4) Line 97 What is mean Tc95?
(5) I think the specific data should be included in the conclusion.
Author Response
Development and Characterization of Unmodified and Modified Natural Rubber Composites Filled with Modified Clay
Adisak Keereerak1, 3, Nusara Sukkhata1, Nussana Lehman1, Yeampon Nakaramontri2,
Karnda Sengloyluan3, Jobish Johns4, Ekwipoo Kalkornsurapranee1*
We thank and respect the comments of reviewers.
In this work, the effect of Et-OH on the dispersion of bentonite clay at various mixing temperatures in case of different type of rubber matrix on dynamic mechanical rheology, Payne effect, XRD and mechanical properties of rubber composites were studied in detail. I think it can be accept after minor revise.
(1) Line 13. “Novel composites based on rubber and modified bentonite clay (Clay) with enhanced dispersion in ethanol solutions (Et-OH) using ultrasonic method was investigated.” This statement is ambiguous, please revise it.
Thank you for the suggestion.
All changes and answers according to reviewer recommendations have been included and highlighted in the text (in the first and second paragraph of abstract).
We changed the sentence to ‘Novel composites based on rubber and modified bentonite clay (Clay) was investigated. The modified bentonite clay was developed by dispersing in ethanol solutions (Et-OH) using ultrasonic method’
(2) I think the author's introduction is not comprehensive in explaining why the intercalation method is used, the role of ethanol in it, and the previous research progress.
The introduction part is modified by including some more points (Page 2) as
It leads them not so suitable to mix and interact with most of the polymer matrices with higher hydrophobicity. Therefore, organically modified layered silicates can be generated by modifying clays with various organic surfactants through intercalation process. Ion exchange is an easy and traditional method to modify the surface of clay. However, the disadvantages of this method including the cations are not strongly bound to the clay surface and also small cation molecules can substitute the cations present in the clay surface [9-10]. The interlayer spacing contains exchangeable cations that can be replaced by organic cationic surfactants (e.g., alkylammonium or alkylphosphonium cations) to improve the compatibility of clay with rubber. This method improves the interlayer distances of montmorillonite and exhibits organophilic and hydrophobic nature on its surface, and the treatment is usually carried out in aqueous or aqueous-ethanol media [11-12].
The clay layer can be steadily dispersed in water for the hydration of ion among the layers; the layers are separated from each other. Some polar compounds can intercalate to the clay layer galleries. These characteristics provide the way to prepare the rubber-clay composites. [13] Intercalation techniques are helpful to insert organic molecule into clay gallery, effectively. Increase of interlayer space in clay particles is the most important factor to disperse clay in rubber matrix. [14-15].
(3) I am very confused about the design of Table 1. I think the content of each component should be stated more clearly. Moreover, I suggest the author define the name of the sample clearly.
Thanks for the suggestion.
We do agree with your recommendation and changed the data in table 1. It is highlighted in table 1.
(4) Line 97 What is mean Tc95?
It has been explained in the text and highlighted as
cure time or vulcanized time (Tc95)
(5) I think the specific data should be included in the conclusion.
The conclusion part has been modified as per the reviewer suggestion.
Rubber composites based on modified bentonite clay dispersed in Et-OH solution by ultrasonic method were successfully prepared. The NR-Clay composite developed by using bentonite clay of 20 phr dispersed in Et-OH at a mixing temperature of 80°C exhibited better properties in terms of Payne effect, XRD analysis and mechanical properties when compared to the composites of unmodified bentonite clay. Based on the effect of rubber types, XRD analysis showed better dispersion in ENR50-Clay composites rather than the other rubber composites. An increase in d-spacing (3.87 to 4.92 nm) and the shift of angle of diffraction to lower angle (19.98 to 19.20 degree) can also be seen from XRD analysis. Moreover, ENR50-Clay composite exhibited better tensile properties than the composites of NR-Clay and also exhibited two times higher than that of ENR25-Clay composite. As a result of increased epoxy content in ENR, the property of ENR matrix was also simultaneously enhanced. On evaluating the performance of composites, ENR50-Clay composite with modified bentonite clay dispersed in Et-OH solution improved the interlayer spacing of bentonite clay (well dispersion) by partial intercalation of rubber matrix.
Reviewer 2 Report
The manuscript reports Development of NR/modified clay composite: effect of novel processing and rubber types on clay dispersion, and I consider that title did not reflect the reported data, I mean the material process has not a novelty compared with previous methods, and rubber types suppose different structure or sources, so I recommend to change the tittle indicating that NR and NR modified were evaluated. I recommend to use NR for code for natural rubber in figures and tables, not Neat. Why decide to not evaluated NR-clay 60ªCly Et in results? I mean if report the processing conditions this material must be reported. Other specific comments are detailed following:
- I recommend to enrich the state of art, due is short, and only 10 references were used for introduction.
-Please define abbreviations written, for instance MBTS.
-For figure 1, please enlarge the font size and also for fig 1b.
-In fig 2, I recommend to include the bentonite XRD pattern.
-For XRD results discussion, please explain what is it mean that peak displace to lower or higher angles. Also, indicate that d-spacing of NR clay composites in EtOH at 80ºC provides highest value, but the difference in minimum (in the order of 0.06 max nm), is this value representative of change?
-Figure 3 and table 3 report the mechanical properties, so I recommend that for avoid duplication of data, delete one of them, I recommend to keep figure 3.
-Table 3 and 4 caption is the same, please correct.
-In line 244 indicate that d-spacing of ENR-clay composites are reported in table 6, but table 6 did not reports those information.
-What relation has Tg value with epoxy content and mechanical properties?, I mean Tg is not reported in this work, so why decide to relate this property with obtained data?
In general, I recommend to improve the manuscript discussion based on comments and change title of work.
Author Response
Development and Characterization of Unmodified and Modified Natural Rubber Composites Filled with Modified Clay
Adisak Keereerak1, 3, Nusara Sukkhata1, Nussana Lehman1, Yeampon Nakaramontri2, Karnda Sengloyluan3, Jobish Johns4, Ekwipoo Kalkornsurapranee1*
We thank and respect the comments of reviewers.
The manuscript reports Development of NR/modified clay composite: effect of novel processing and rubber types on clay dispersion, and I consider that title did not reflect the reported data, I mean the material process has not a novelty compared with previous methods, and rubber types suppose different structure or sources, so I recommend to change the tittle indicating that NR and NR modified were evaluated. I recommend to use NR for code for natural rubber in figures and tables, not Neat. Why decide to not evaluated NR-clay 60ªCly Et in results? I mean if report the processing conditions this material must be reported. Other specific comments are detailed following:
The manuscript has been modified as per your valuable suggestions. Neat term is changed to NR in the manuscript.
(1) I recommend to enrich the state of art, due is short, and only 10 references were used for introduction.
Thanks for your suggestion. We have added the following points in introduction (Page 2).
It leads them not so suitable to mix and interact with most of the polymer matrices with higher hydrophobicity. Therefore, organically modified layered silicates can be generated by modifying clays with various organic surfactants through intercalation process. Ion exchange is an easy and traditional method to modify the surface of clay. However, the disadvantages of this method including the cations are not strongly bound to the clay surface and also small cation molecules can substitute the cations present in the clay surface [9-10]. The interlayer spacing contains exchangeable cations that can be replaced by organic cationic surfactants (e.g., alkylammonium or alkylphosphonium cations) to improve the compatibility of clay with rubber. This method improves the interlayer distances of montmorillonite and exhibits organophilic and hydrophobic nature on its surface, and the treatment is usually carried out in aqueous or aqueous-ethanol media [11-12].
The clay layer can be steadily dispersed in water for the hydration of ion among the layers; the layers are separated from each other. Some polar compounds can intercalate to the clay layer galleries. These characteristics provide the way to prepare the rubber-clay composites. [13] Intercalation techniques are helpful to insert organic molecule into clay gallery, effectively. Increase of interlayer space in clay particles is the most important factor to disperse clay in rubber matrix. [14-15].
(2) Please define abbreviations written, for instance MBTS.
The expansion of abbreviation has been included below the table.
(3) For figure 1, please enlarge the font size and also for fig 1b.
Edited as per the suggestion
(4) In fig 2, I recommend to include the bentonite XRD pattern.
It has been edited by including XRD analysis of bentonite clay for more explanation in the discussion.
(5) For XRD results discussion, please explain what is it mean that peak displace to lower or higher angles. Also, indicate that d-spacing of NR clay composites in EtOH at 80ºC provides highest value, but the difference in minimum (in the order of 0.06 max nm), is this value representative of change?
We do agree with this and edited the as shown in the highlighted texts.
Figure 3 illustrates the XRD patterns of NR and NR-Clay composites with and without dispersing bentonite clay in Et-OH solution at different mixing temperatures. Pure NR shows clearly the amorphous characteristic as it does not have any crystalline peaks. Normally, the diffraction peaks can be seen at the angles 5.71°, 7.61°, 12.55°, 19.89°, 20.53°, 21.03°, 21.42°, 25.09°, 35.05°, 35.92° and 61.69° [19]. The NR-Clay composites prepared at the mixing temperature of 60°C exhibits the characteristic peaks at the angle of diffraction 2θ at 11.28°, 22.86°, 27.86°and 28.81°. The NR-Clay composites developed at the mixing temperature of 80°C show some sharp peaks at 10.12°, 22.60°, 26.88° and 28.33°. The diffraction angle is related to the characteristic peaks of bentonite clay at 21.42° (110), 27.20° (210) and 30.01° (124) [19-21]. Moreover, the XRD patterns indicated that the NR-Clay composites consist of Heulandite-Ca and Montmorillonite-15A as the major and minor constituents.
In addition, the angle of diffraction displaced towards lower value upon adding modified bentonite clay in the composites. Especially, the angle reduces from 19.94° to 18.15° that shows the increment of inter-planar spacing of 3.85 to 4.99 nm. From these results, the d-spacing of NR-Clay composites with bentonite clay in Et-OH and temperature of 80°C provides the highest value among the other NR-Clay composites as shown in table 2. This is the reason to exhibit better homogeneous dispersion of bentonite clay in NR-Clay composite using bentonite clay dispersed in Et-OH before mixing in internal mixer. Moreover, the increasing of inter-planar spacing reveals that the intercalation of NR molecular chains in between the layers of bentonite clay. It is an indication of enhanced mechanical properties compared to the NR-Clay composites prepared without dispersing in Et-OH. Moreover, these results are related to the values of Payne effect and it can be explained in terms of lower filler-filler interaction.
(6) Figure 3 and table 3 report the mechanical properties, so I recommend that for avoid duplication of data, delete one of them, I recommend to keep figure 3.
Agreed the suggestion to keep figure 3. But the figure 3 becomes figure 4 after edition.
(7) Table 3 and 4 caption is the same, please correct.
Deleted table 3. So, caption (in table 4) no need to change table name. But the table become table 4 after edition.
(8) In line 244 indicate that d-spacing of ENR-clay composites are reported in table 6, but table 6 did not reports those information.
The discussion part has been changed as per the suggestion of reviewer and highlighted in the text below table 4 (page 12).
Figure 9 shows the XRD patterns of NR-Clay, ENR25-Clay and ENR50-Clay composites. The ENR25-Clayet composites exhibited peaks at the diffraction angles 22.94°, 26.8° and 28.13°. Pointed peaks are observed at 22.71°, 26.42° and 28.15° in case of ENR50-Clayet composites. In cases of NR-Clay, ENR25-Clay and ENR50-Clay composites, the broad peak in the range of 12-24 degree shifts to lower side. And also the angles slide to lower the angle from 19.94, 19.91 and 19.42, respectively. Additionally, the ENR50-Clay composite shows the displacement of peaks at 10.03° and 29°. This is attribute to the impregnation of elastomeric phase into the silicate galleries, hiding the regular stacked-layer structure of the modified bentonite clays and extends to an intercalated structure. Moreover, intensity of the sharp peak (characteristic peak) is also higher on increasing the epoxy degree in ENR composites. Dominantly, the intensity of diffraction peak at 10.03° is slightly higher than 9.99 and 9.9°. The d-spacing is found to be increased from 3.87 to 4.92 nm at 19.98°, 19.91° and 19.20° as shown in table 4. It exhibits the intercalation of rubber molecules in between the layers of bentonite clay. Besides, the polarity of the modified NR (ENR) supports intercalation of elastomer chain into the modified bentonite clay leads to better intercalation and dispersion in the matrix [23, 24].
(9) What relation has Tg value with epoxy content and mechanical properties?, I mean Tg is not reported in this work, so why decide to relate this property with obtained data?
Changed the discussion as per the reviewer recommendations and highlighted the text (page 12).
The Tg of ENR25 and ENR50 are -47.5°C and -30°C, respectively. Moreover, the Tg of ENR increases linearly with the degree of epoxidation or epoxy content [24]. From previous research, Yokkhun et al. (2014) studied the preparation and characterization of ENR/modified montmorillonite clay (OC-MMT) composite. The Tg was found to be increased from -28°C up to 18°C along with epoxy content (10-50%). Increment of tensile properties was observed as a result of strong interaction between ENR and OC-MMT and also the degree of strain-induced crystallinity in ENR matrix [25].
Round 2
Reviewer 2 Report
After review the corrected version, this shows a significant improve, so I can recommend to ACCEPT IN PRESENT FORM